# Restoring Shank3 in the rostral brainstem of *shank3ab−/−* zebrafish autism models rescues sensory deficits

Robert A. Kozol [1,3✉], David M. James[1,4,5], Ivan Varela[1], Sureni H. Sumathipala[1], Stephan Züchner [2] & Julia E. Dallman [1✉]

People with Phelan-McDermid Syndrome, caused by mutations in the *SHANK3* gene, commonly exhibit reduced responses to sensory stimuli; yet the changes in brain-wide activity that link these symptoms to mutations in the *shank3* gene remain unknown. Here we quantify movement in response to sudden darkness in larvae of two *shank3* zebrafish mutant models and show that both models exhibit dampened responses to this stimulus. Using brain-wide activity mapping, we find that *shank3−/−* light-sensing brain regions show normal levels of activity while sensorimotor integration and motor regions are less active. Specifically restoring Shank3 function in a sensorimotor nucleus of the rostral brainstem enables the *shank3−/−* model to respond like wild-type. In sum, we find that reduced sensory responsiveness in *shank3−/−* models is associated with reduced activity in sensory processing brain regions and can be rescued by restoring Shank3 function in the rostral brainstem. These studies highlight the importance of Shank3 function in the rostral brainstem for integrating sensory inputs to generate behavioral adaptations to changing sensory stimuli.

[1] Department of Biology, University of Miami, Coral Gables, FL, USA. [2] Dr. John T. Macdonald Foundation Department of Human Genetics and John P. Hussman Institute for Human Genomics, University of Miami, Miami, FL, USA. [3] Present address: Jupiter Life Science Initiative, Florida Atlantic University, Jupiter, FL, USA. [4] Present address: Knight Campus for Accelerating Scientific Impact, University of Oregon, Eugene, OR, USA. [5] These authors contributed equally: Robert A. Kozol, David M. James. ✉email: rkozol@my.uri.edu; j.dallman@miami.edu

Altered sensory processing is a pervasive but poorly understood symptom in individuals with autism spectrum disorders (ASD)[1]. Sensory symptoms manifest as dampened or excessive responses to light, sound, and/or touch. Because of variability in both the presence and presentation of sensory symptoms, gaining a mechanistic understanding of these sensory processing deficits remains a challenge. In contrast to ASD as a whole, genetically defined forms of ASD share similar sensory deficits. For instance, individuals with Phelan McDermid Syndrome (PMS), a syndromic form of ASD, show low sensitivity to pain and reduced responses to auditory and visual stimuli[2,3]. PMS is caused by the loss of function of one copy of the SHANK3 gene, due to either terminal deletions of chromosome 22[4] or SHANK3 point mutations[2]. In this study, we identify rostral brainstem as a region that requires Shank3 function for normal behavioral responses to sudden darkness in zebrafish models of PMS.

Several animal models of PMS recapitulate dampened responses to diverse sensory stimuli: pain in Shank3 mutant mice[5], sound in Shank3 mutant rats[6], and both touch and light in shank3ab mutant zebrafish[7,8]; nonetheless, a brain-wide understanding of these dampened responses is lacking. Hyperactivity in PMS could reflect functional changes that either span the entire brain or are localized to specific brain regions and/or muscle[9]. Zebrafish allow unique experimental approaches to identify underlying mechanisms because, within the first week of life, larvae have fully functional sensory-motor circuits and produce robust, stereotyped responses to calibrated sensory stimuli[10]. These larval zebrafish have transparent vertebrate brains composed of only ~100,000 neurons, allowing unbiased functional approaches to map brain-wide neuronal activity[11,12]. Moreover, embryonic transplantation can be used to make wild-type-mutant chimeras to test for brain-region-specific functional rescue[13]. Here, we use brain-wide activity mapping and transplants to identify and functionally validate brain regions that underlie sensory hyperreactivity to sudden darkness in zebrafish shank3 mutant models.

## Results and discussion
In contrast to the single SHANK3 gene in people, the shank3 gene is duplicated in zebrafish; therefore, to generate zebrafish models of PMS we used CRISPR/Cas9 to mutate both the shank3a and shank3b (shank3ab) gene paralogs. Shank3 proteins are large, ~200 kD, with multiple isoforms that can be differentially impacted by mutations in different parts of the gene[14]. To capture this complexity, we generated two zebrafish PMS models, shank3abΔN with mutations truncating both the Shank3 a and b proteins in the ankyrin repeat domains and shank3abΔC with mutations truncating both the Shank3 a and b proteins near the proline-rich domain[14] (Fig. 1a; Supplementary Fig. 1, Supplementary Data 1). These models mimic the most common types of SHANK3 mutations found in people with PMS[2] and, by having two models, we control for genetic background. In mice and humans, Shank3 protein is expressed in glutamatergic granule cells of the cerebellum, colocalizing with the scaffolding protein PSD-95. Likewise, in wild-type zebrafish, we show that Shank3 protein colocalizes with PSD-95 in the cerebellum and along ventral neural tracts of the brainstem (Fig. 1b; Supplementary Fig. 2). In contrast, in both shank3abΔN−/− and shank3abΔC−/− PMS models Shank3 puncta are lacking despite intact PSD-95 synaptic puncta (Fig. 1b). These data indicate that the four alleles that underlie the two shank3abΔN−/− and shank3abΔC−/− models are loss-of-function mutations. Hereafter, we refer to shank3abΔN−/− and shank3abΔC−/− models are as shank3ab−/− models except in cases that the results differ between the models.

The sensory reactivity of zebrafish shank3ab−/− models was measured by quantifying behavioral changes to a light-based stimulus using the well-established visual motor response (VMR;[15]). The VMR is characterized by dramatic increases in movement in response to sudden transitions from light to darkness (Fig. 1c). Both shank3ab−/− models exhibited reduced VMR responses as quantified by comparing the distance traveled in the 30 s before and after transitions from lights-on to lights-off conditions (Fig. 1c–e; Supplementary Data 1, Sheets 1–6). Dampened VMR responses were more pronounced in homozygous shank3ab−/− larvae ($p < 0.001$) than in heterozygous shank3ab +/− larvae ($p < 0.05$). In comparison to the VMR response, lights-on baseline locomotor activity was more variable across trials in both wild type and shank3ab mutant larvae (Supplementary Fig. 3, Supplementary Data 1, Sheets 7–15). We used the pronounced VMR deficits in shank3ab−/− mutants as the basis of all subsequent experiments to determine the mechanistic underpinnings of these altered sensorimotor integration phenotypes.

To identify the neural circuits underlying hyporeactivity in shank3ab mutant models, we used an unbiased, brain-wide, Mitogen-Activated Protein (MAP)-mapping[12] approach, based on phosphorylation of extracellular signal-regulated kinase (pERK). Because ERK phosphorylation increases when calcium is elevated during action potentials, staining for pERK provides a proxy for neuronal activity (Fig. 2a, b). Brain regions differentially active between light-on and lights-off conditions were identified by statistically comparing relative ERK signals (pERK/total ERK) in two groups of 15–21 larvae per group ($p < 10^{-5}$; Fig. 2c, d, Supplementary Fig. 4). In response to the lights-on stimulus, wild type (WT) and shank3ab−/− models showed similarly elevated pERK staining in the optic tectum (green) that receives input from retinal ganglion cells. In response to the lights-off stimulus, WT showed elevated pERK staining in the pineal, the telencephalic pallium and subpallium, the torus semicircularis of the midbrain, brainstem, and spinal cord (magenta). While both N- and C- shank3ab−/− mutant models showed similarly elevated pERK staining in the pineal, they showed little or no elevated pERK in other brain regions. These VMR brain activity maps in shank3ab−/− models show that sensory brain regions including the pineal, retina, and optic tectum detect changes in light normally, but that downstream brain regions fail to integrate and respond to dark transitions consistent with dampened lights-off behavioral responses.

Next we explored whether restoring Shank3 function would be sufficient to rescue hyporeactivity in both shank3abΔN−/− and ΔC−/− models. We generated genetically mosaic larvae by transplanting WT cells into otherwise shank3ab mutant embryos at the late gastrula shield stage, ~6 h post-fertilization (Fig. 3a). WT donor cells were deposited in the region of the shank3ab−/− embryo fated to become brainstem. To track the fate of transplanted cells, WT donor Zebrabow embryos expressing dTomato under a ubiquitin promoter[16] were used as the source of WT cells, referred to as Zb-T for Zebrabow transplant (Fig. 3; Supplementary Fig. 5). Remarkably, when tested as six-day-old larvae, transplanted Zb-T cells were sufficient to rescue shank3ab−/− mutant lights-off reactivity in the VMR assay (Fig. 3b–f, Supplementary Fig. 5 ZB-T & 6 WT-T; Supplementary Data 1, Sheets 16-23). To determine Zb-T brain regions in common among behaviorally rescued shank3abΔ:Zb-T larvae, we registered shank3abΔ:Zb-T larvae to the Z-brain atlas. We found that the majority of rescued shank3abΔ:Zb-T larvae had integrated Zb-T cells in a rostral dorsal glutamatergic brainstem nucleus referred to in the Z-brain atlas as vGluT cluster 2 (90.5%; $n = 19/21$; Supplementary Fig. 7; Supplementary Data 1, Sheet 24). To control for non-specific transplantation effects, we performed

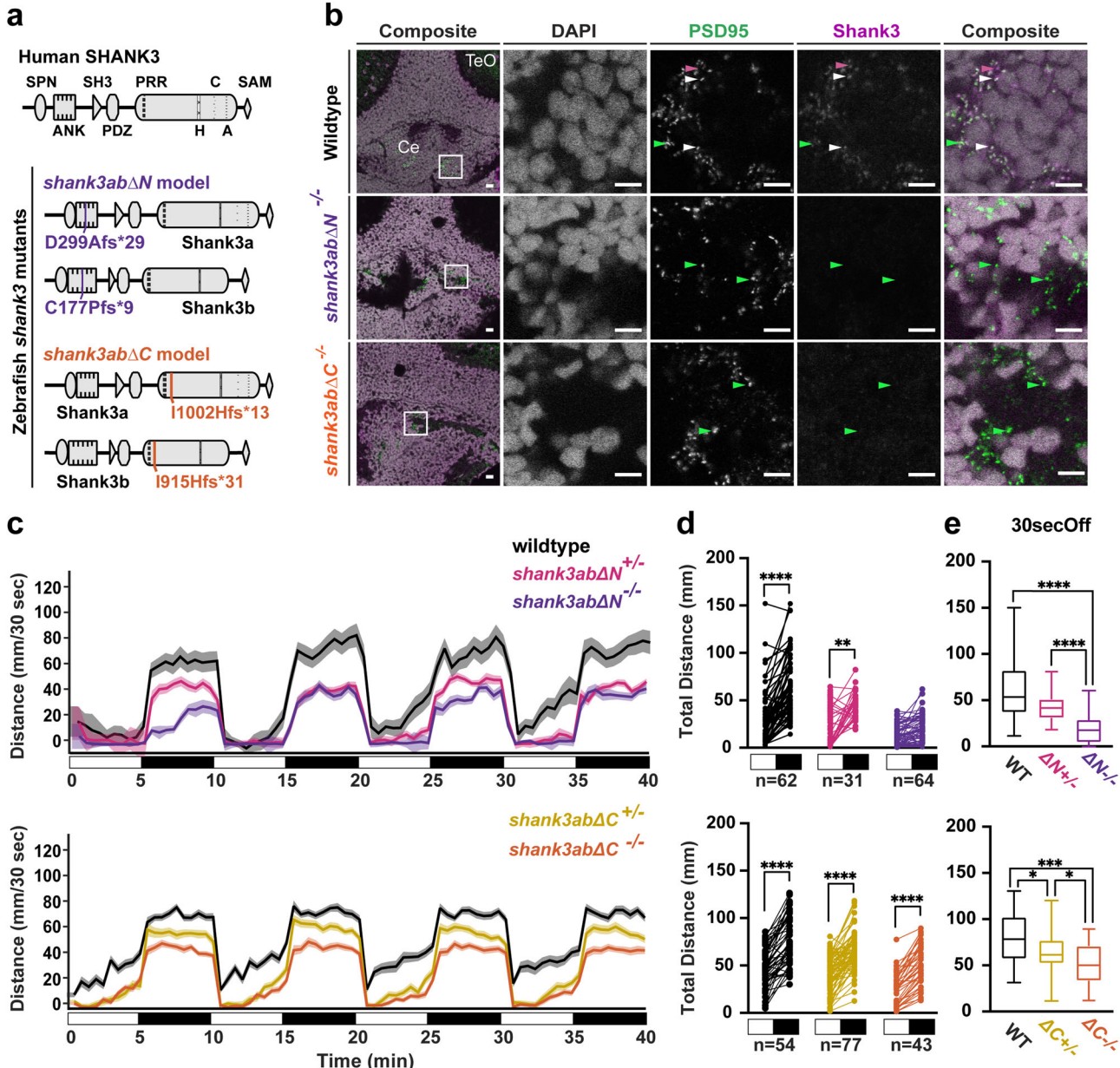

**Fig. 1 Two independent shank3ab mutant models are hyporeactive to lights-off stimuli. a** Shank3 protein diagrams of human SHANK3 and duplicated zebrafish Shank3a and Shank3b show where in the zebrafish proteins four independent CRISPR-Cas 9 indel alleles introduce frameshift mutations. Protein interaction domains indicated in human SHANK3 are more highly conserved in zebrafish Shank3a than Shank3b (SPN = Shank/ProSAP N-terminal, ANK = ankyrin repeats, SH = SRC Homology 3, PDZ = post-synaptic density protein/disc large/zonula occludens-1, PRR = proline-rich region that includes interaction domains with H = Homer, C = cortactin, A = actin binding protein 1, and SAM = sterile alpha motif). Each *shank3abΔN* (purple) and *shank3abΔC* (orange) mutant model has similar mutations in Shank3a and Shank3b paralogs: *shank3abΔN* mutations are in ankyrin repeat regions and *shank3abΔC* mutations are in the proline-rich region. **b** Coronal cryosections from 6dpf larvae were stained with antibodies against the glutamatergic post-synaptic scaffolding proteins PSD-95 and Shank3, with a representative sample shown for each genotype. Synapses in wild-type cerebellum (Ce) stain for both PSD-95 (green arrowheads) and Shank3 (magenta arrowheads) puncta, some of which colocalize (white arrowheads), compared to *shank3ab* mutants that stain for PSD-95, but not Shank3. TeO= Optic Tectum; Scale bars represent 10 μm. **c** Visual motor responses (VMR) are shown as line graphs of median distance traveled in 30 s ±SE to four cycles of lights-on to lights-off transitions. White and black boxes below the *x*-axis indicate alternating lights-on and lights-off, respectively. Exact sample sizes of biologically independent samples are indicated below the Paired dot plots and apply to plots in (**c**–**e**). **d** Paired dot plots compare median swimming distances per larva of the four light transitions in the 30 s before and after the lights-on to lights-off transition. Within genotype comparisons were conducted using Dunn-Bonferroni *p*-value corrected t-tests (**e**) Box plots compare distance traveled during the first 30 s of dark between WT and *shank3ab* mutant models. Boxes denote the median, 1st and 3rd quartile, while whiskers represent the minimum and maximum values. Groups were statistically compared using a Kruskal–Wallis ANOVA, and when $p < 0.05$, were followed by a Dunn's multiple comparison test. *P*-value asterisks represent; $p < 0.05$ - *, $p < 0.01$ - **, $p < 0.001$ - ***, $p < 0.0001$-****. Source data for plots are provided in Supplementary Data 2.

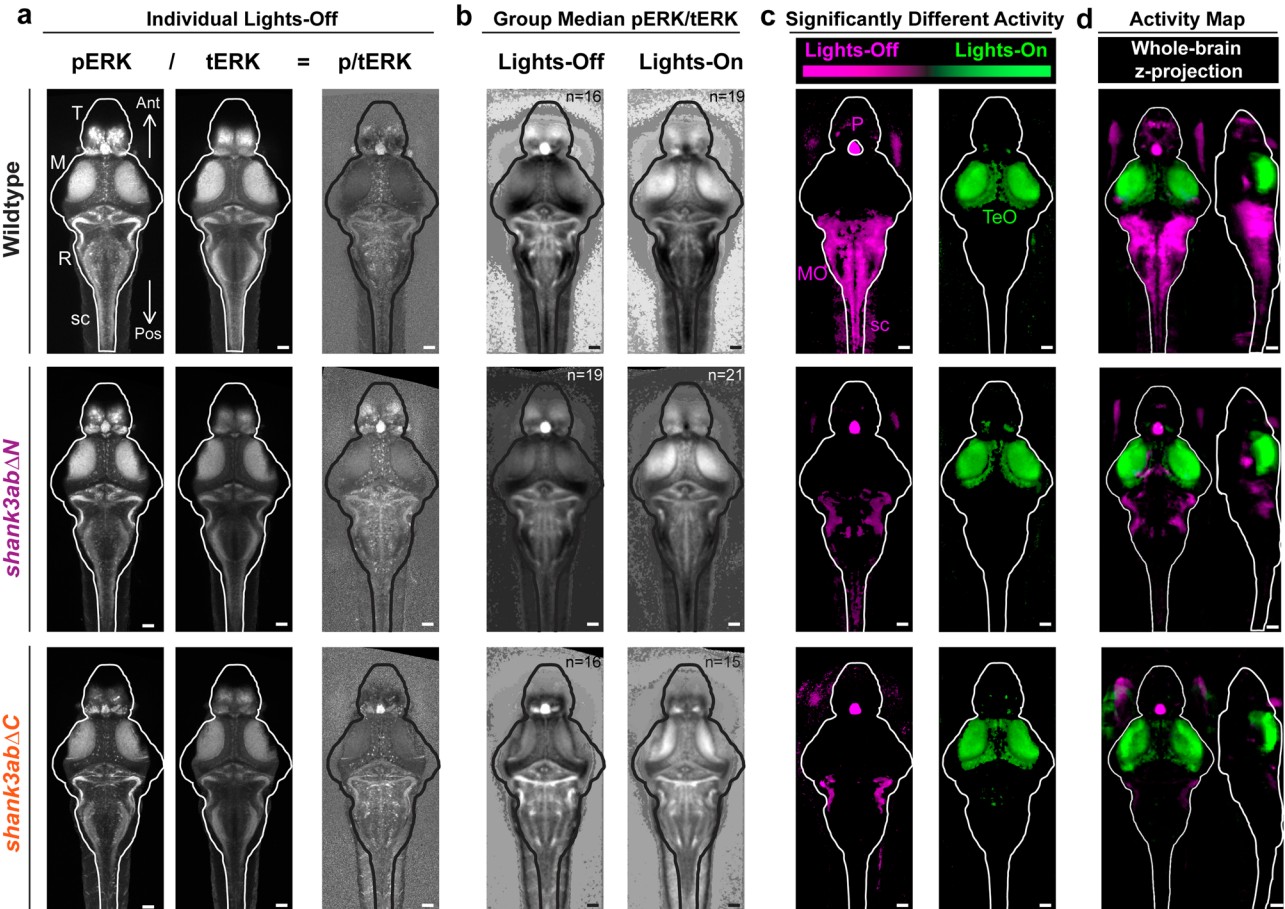

**Fig. 2 Brain-wide neural activity mapping reveals shank3abΔ−/− mutant models sense light normally but fail to activate downstream brain regions underlying sensorimotor integration.** Brain-wide activity maps were generated by using phosphorylated-ERK (pERK) antibody staining as a proxy for neuronal activity. **a** Individual larval stacks were registered for use with the Z-brain atlas and MAP-mapping MATLAB scripts (Randlett et al. [12], Engert lab) Individual pERK stacks left were then divided by total-ERK (tERK; middle), providing normalized pERK/tERK signal (right). **b** Median p/tERK values were then calculated for every voxel within the brain for each genotype and light condition (Exact sample sizes of biologically independent samples for each condition and genotype (n = lights-on/lights-off); wild-type (n = 16/19), shank3abΔN (n = 19/21) and shank3abΔC (n = 16/15) are indicated on group median images in (**b**) and also apply to (**c**) and (**d**). **c** Mann–Whitney U z-scores were calculated, comparing lights-off and lights-on, with magenta indicating increased activity during the transition to lights-off (e.g. Medulla Oblongata, MO) and green indicating increased activity during the transition to lights-on (e.g. Optic Tectum, TeO). Regions within the brain that are black did not reach the $P < 10^{-5}$ cut-off. **d** In comparison to wild-type, shank3abΔ−/− mutant models respond to the lights-off condition (magenta) with activation of their pineal (P), but fail to show activation in the MO and spinal cord (sc). **a–c** All images are 20 μm dorsal z-projections. **d** Whole brain z- and x-projections. Scale bars = 50 μm.

within genotype transplants. WT donor to WT recipient chimeras and shank3abΔN−/− donor to shank3abΔN−/− recipient chimeras had no effects on VMR behaviors compared to unmanipulated larvae of the corresponding genotype. shank3abΔC−/− donor to shank3abΔC−/− recipient chimeras had more severe hyporeactivity compared to unmanipulated larvae of the same genotype (Supplementary Figs. 8, Supplementary Data 1, Sheets 25–30). Consistent with the MAP-mapping experiments, these results indicate Shank3ab function in rostral brainstem is sufficient for WT levels of light-evoked activity.

To gain a better understanding of this rostral brainstem region, we first used zebrafish atlases to determine local anatomy, and functional studies for response characteristics. vGlut2 cluster 2 is the most rostral and dorsolateral glutamatergic nucleus [17,18] of a part of the Medulla Oblongata that is derived from rhombomeres three and four (Z-brain atlas [12]; ZBB Zebrafish Brain Browser [19]). This region encompasses the rostral portion of the Medial Octavolateral Nucleus (MON), a large nucleus that spans rhombomeres three through six [19]. Studies using electrophysiology and whole-brain GCaMP have identified this rostral

MON nucleus as important in transforming sensory stimuli, including water flow [18,20,21], visual [22], and sound [23–25], into behavioral responses. The MON is considered "cerebellar-like," with glutamatergic cells that form a mass (rather than a layer in the cerebellum), and with circuits set up to evaluate changes in multimodal sensory stimuli that arise from the movement of the body as compared to changes in the environment [26]. Consistent with this, functional studies have shown that activity in the rostral MON correlates with behavioral adaptations to changes in sensory stimuli [22,25].

We suggest that a related "cerebellar-like" circuit in mammals is the Cochlear complex, though the arrangement and anatomy of these nuclei has changed with evolution of the cochlea, associated with the transition from water to land [27]. Like the MON in zebrafish, the Cochlear complex consists of dorsolateral nuclei derived from mouse atonal homolog 1(Math1)-expressing cells of rhombomeres three and four [28,29]. Disruptions to rhombomere three, Math1-expressing cells in mice results in profound hearing loss as measured by auditory brainstem recordings [30], demonstrating its critical role in central auditory processing. Moreover,

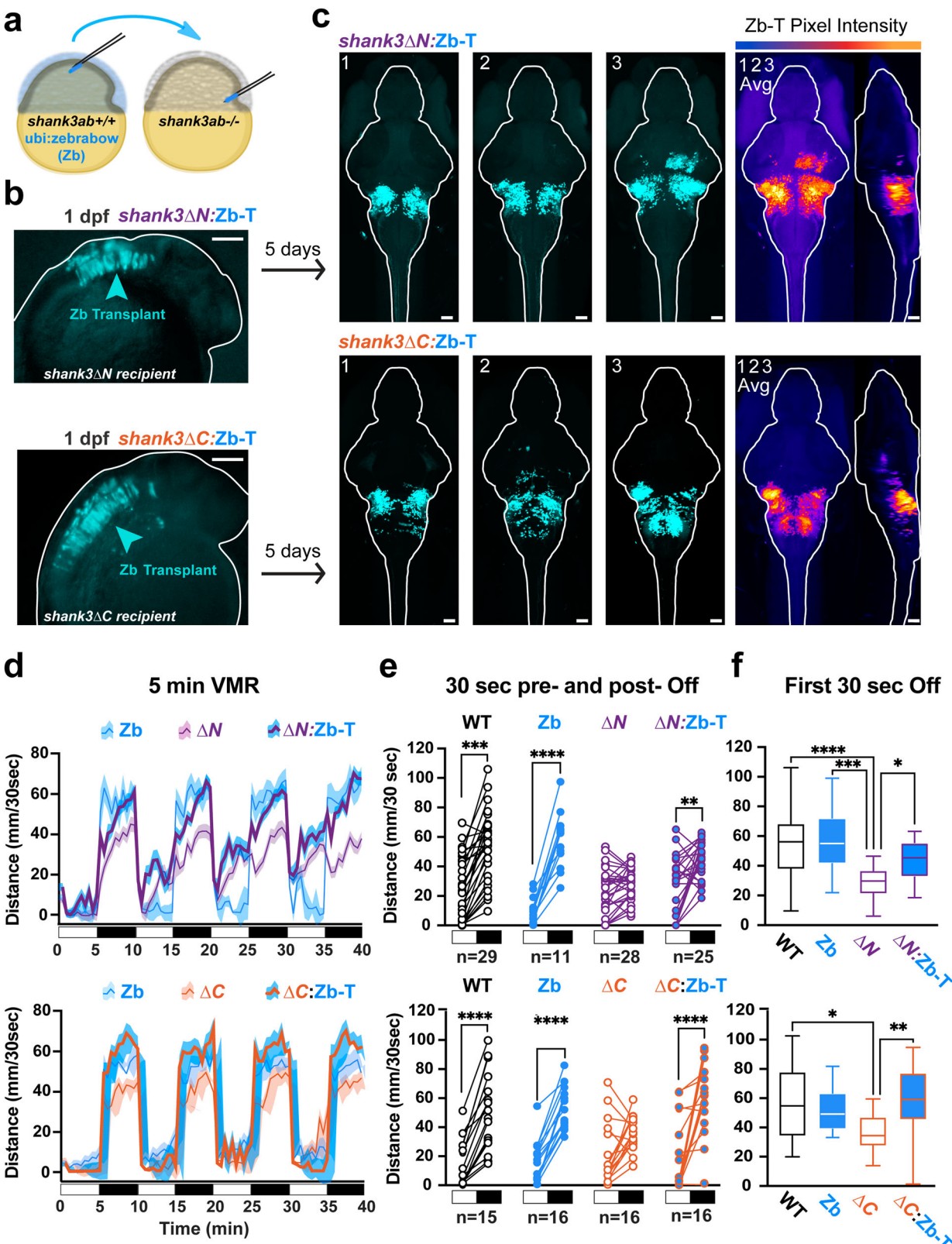

the cerebellar-like structure of this brain region is thought to mediate predictions of how body movement will impact multi-sensory integration[26]. Our findings suggest that Shank3 function in these brainstem nuclei plays an important role in sensorimotor integration.

Brainstem deficits in *shank3abΔN−/−* and *ΔC−/−* mutants could be due to synaptic deficits and/or altered development. In

support of a synaptic role, loss of Shank3 protein in mammalian models is known to decrease glutamate receptor expression, disrupt post-synaptic density composition, and reduce synaptic transmission[31]. Weaker excitatory synaptic responses could therefore explain the failure of sensory brain regions to evoke responses at the levels of both other brain regions and motor behaviors in *shank3ab* mutant PMS models.

**Fig. 3 Hyporeactivity is rescued in both ΔN and ΔC *shank3* mutant models by restoring wild-type shank3ab positive neurons in dorsal/rostral glutamatergic brainstem nuclei. a** A cartoon shows how cells from wild type donor embryos marked by a ubiquitously expressed dTomato fluorescent protein (ubi:zebrabow) are transplanted into the presumptive hindbrain of *shank3ab−/−* mutant recipient embryos at mid-gastrulation stages. **b** Chimeric embryos at 1 day post-fertilization (dpf), with donor cells expressing the fluorescent protein (false-colored in cyan) in recipient *shank3abΔN−/−* or *shank3abΔC−/−* embryos. Chimeric six-day-old larvae (*shank3ab−/−*:Zb-T) were imaged to determine the fate of the transplanted cells. **c** Confocal images of chimeric larvae at 6 dpf following behavioral screening, demonstrating transplanted cells in rescued larvae populate the dorsal/rostral brainstem nuclei. Individual representative larvae are numbered 1-3, with the three averaged in the right most stack. **d** VMR line graphs, median +/− SE, (**d–f**) and (**e**) paired dot plots show lights-off behavioral phenotypes are rescued in both *shank3abΔ−/−* mutant models with wild-type-derived brain stems (*shank3abΔ−/−*:Zb-T). Exact sample sizes of biologically independent samples for each genotype and chimera are indicated below each plot and also apply to d and f. Within *shank3* model comparisons were conducted using Dunn–Bonferroni *p*-value corrected t-tests. **f** Box plots displaying median swimming distances for individuals following the first 30 s following lights-off. Individual values are medians representing all four lights-off transitions for individual larvae. Boxes denote the median, 1st and 3rd quartile, while whiskers represent the minimum and maximum values. Groups were statistically compared using Kruskal-Wallis one-way ANOVA, and when $p < 0.05$, were followed by Dunn's multiple comparisons. *P*-value asterisks represent; $p < 0.05$ - *, $p < 0.01$ - **, $p < 0.001$ - ***, $p < 0.0001$-****. Scale bars = 100 μm (**b**); 50 μm (**c**). Source data for plots are provided in Supplementary Data 2.

Functional deficits could also be due to altered development that could disrupt functional connectivity. Supporting this possibility: global developmental delay has previously been reported in *shank3ab* zebrafish models[7,8]; *shank3ab* transcripts are expressed throughout embryonic development, prior to synaptogenesis[7]; and *shank3* has been implicated in wnt signaling[32,33]. Moreover, altered brainstem development has been suggested as the likely basis for multisensory integration and sensory-motor deficits more generally in ASD[34,35]. To further explore this issue, we compared brain morphology between wild-type and *shank3* mutants using CobraZ, an anatomical measurement tool[36]. CobraZ segments the zebrafish brain into 180 neuroanatomical regions, then computes the pixels of each brain region as a percentage of total brain pixels. To make these brain regions more clinically relatable, we reduced the ZBB atlas from 180 brain segments to 26 larger regions, with clear equivalents in human brain (Table 1). In both *shank3ab* mutant models, the dorsal diencephalon, valvula cerebelli and locus coeruleus were larger while the medulla oblongata was smaller (Fig. 4 and Supplementary Fig. 9). Furthermore, *shank3*:Zb-T transplants rescued anatomical scaling of both the locus coeruleus and medulla oblongata (Fig. 4a, b, Supplementary Data 1, Sheets 31–38). These findings suggest that both neurodevelopmental and synaptic deficits could contribute to altered sensory processing in *shank3* mutant zebrafish. With the recent inclusion of sensory deficits, more clinical research is needed to determine links between changes in brainstem function and sensory deficits in individuals with autism.

In summary, we used whole-brain activity mapping and chimeric rescue experiments to identify the rostral brainstem as a region that requires Shank3 function to generate behavioral responses to visual stimuli. To do this we generated two independent zebrafish *shank3abΔN* and *shank3abΔC* mutant models of Phelan McDermid Syndrome, both of which, like humans with Phelan McDermid Syndrome, showed dampened responses to visual stimuli. Unbiased, whole-brain activity mapping in both *shank3ab* models was consistent, demonstrating that regions of the brain that detect light were activated normally while brain regions that integrate sensory information to produce specific motor responses showed reduced activation. Restoring Shank3 in brainstem nuclei by transplanting WT cells into *shank3ab* mutant embryos at gastrula stages was sufficient to rescue dampened larval sensory responses, demonstrating a critical role of Shank3 for integrating sensory information in the rostral brainstem. An analysis of brain regions that could rescue sensorimotor behaviors in *shank3ab* models supports an essential role for shank3 in these cerebellar-like brainstem circuits.

| Table 1 Atlas key 26 segment CobraZ atlas. | | |
|---|---|---|
| **Region** | **Pixel code** | **Color** |
| Pallium | 1 | Cyan-blue |
| Subpallium | 2 | Vermilion |
| Cerebellum | 3 | Green-cyan |
| Optic Tectum Neuropil | 4 | Blue-magenta |
| Optic Tectum Grey Matter | 5 | Green |
| Thalamus | 6 | Cyan |
| Rostral Zone (Hypothalamus) | 7 | Dark magenta |
| Intermediate Zone (Hypothalamus) | 8 | Light blue |
| Caudal Zone (Hypothalamus) | 9 | Neon green |
| Pons | 10 | Dark purple |
| Prepontine | 11 | Forest green |
| Reticulopontine | 12 | Green-blue |
| Posterior Tuberculum (Diencephalon) | 13 | Light red |
| Statoacoustic Ganglion (Brainstem) | 14 | Light purple |
| Tegmentum | 15 | Yellow-green |
| Locus Coeruleus | 16 | Light green |
| Preoptic area | 17 | Magenta |
| Adjacent to Posterior Tuberculum (Diencephalon) | 18 | Mustard |
| Optic Tract | 19 | Light sky blue |
| Nucleus Medial Longitudinal Fasciculus | 20 | Light crimson |
| Torus Semicircularis | 21 | Pink purple |
| Vagal Ganglia | 22 | Yellow |
| Dorsal Diencephalon | 23 | Pink |
| Medulla Oblongata | 24 | Orange |
| Raphe Nuclei | 25 | Light orange |
| Valvula Cerebelli | 26 | Beige |

Colors defined by 2-2-4 LUT color scheme in Fiji.

## Conclusion

Brain-wide activity mapping and transplant rescue experiments provide robust evidence that hyperreactivity to light-based stimuli in zebrafish *shank3ab* mutants is due to functional deficits downstream of sensory reception that can be rescued by restoring wild type Shank3 in cerebellar-like circuits of the rostral brainstem.

## Methods

**Fish maintenance and husbandry.** Zebrafish were housed in the University of Miami zebrafish core facility. Both adult and larval zebrafish were maintained at 28 °C in system water and exposed to a 14:10 h circadian light:dark cycle. All experiments were carried out on larvae prior to sexual maturation. Zebrafish were cared for in accordance with NIH guidelines and all experiments were approved by the University of Miami Institutional Care and Use Committee protocol #'s 15–128 and 18–128.

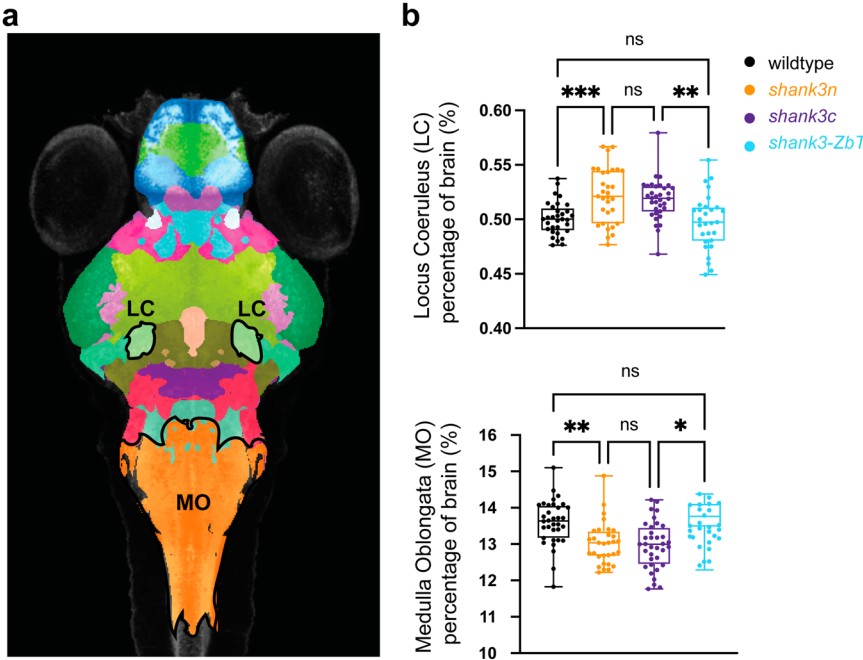

**Fig. 4 Brain regions with volumetric differences in *shank3ab* mutants that were rescued by wild-type transplanted brainstem. a** z-stack image from CobraZ reference and 26 segment atlas. Regions outlined in black denote the locus coeruleus (LC) and medulla oblongata (MO). Scale bar = 100 µm. **b** Box plots comparing relative volume of LC and MO brain segments in wild-type, *shank3abΔn* and *shank3abΔc* mutants, and *shank3ab* mutants with wild-type transplanted brain stems (*shank3abΔZb-T*). Exact sample sizes of biologically independent samples for each condition and genotype (n = lights-on/lights-off); wild-type (n = 16/19), *shank3abΔN* (n = 19/21) and *shank3abΔC* (n = 16/15). Boxes denote the median, 1st and 3rd quartile, while whiskers represent the minimum and maximum values. Brain segment sizes were analyzed using a non-parametric Kruskal–Wallis one-way ANOVA and followed by a Dunn's corrected multiple values comparison. P-value asterisks represent; $p < 0.05$ - *, $p < 0.01$ - **, $p < 0.001$ - ***, $p < 0.0001$-****. Source data for plots are provided in Supplementary Data 2.

**shank3abΔN CRISPR-Cas9 mutant generation**. Zebrafish used for this study include AB-TL wild-type, *shank3abΔN* and *shank3abΔC*[14]. To generate *shank3abΔN* mutants, DNA oligos (see Supplementary Data 1) targeting exon 6 of *shank3a* and exon 4 of *shank3b* were designed using the CHOPCHOP online software tool[37]. To generate DNA templates for in vitro transcription of sgRNAs, gene-specific oligos were annealed to complimentary ends in the universal tracRNA and then overhangs were filled in using high-fidelity DNA polymerase: 1× transcription buffer, 1 µM site-specific oligo, 1 µM tracRNA oligo, 500 nM dNTPs, 0.5 U Phusion high-fidelity DNA polymerase (New England Biolabs, NEB; Ipswich, MA), and nuclease-free H₂O to 10 µL. Reactions were then heated to 95 °C for 1 min and then cooled (0.1 °C/s) to 52 °C prior to incubating at 72 °C for 10 min. Templates were used to synthesize sgRNAs using the MEGAscript (Ambion, Foster City, CA) in vitro transcription kit following kit instructions. RNAs were cleaned using ammonium acetate/ethanol precipitation, resuspended, and aliquoted for injections. For CRISPR/Cas9 injections, sgRNA and Cas9 (PNA Bio; #CP01) were mixed at a 4:1 ratio by mass and incubated at 37 °C for 5 min prior to loading into injection needles. Injection volumes were calibrated to deliver 400 pg:100 pg sgRNA:Cas9 per fertilized zygote. Embryos were raised for 24 h before preparing genomic DNA to check for mutagenic efficiency using primers listed in Supplementary Data and Sanger sequencing. F₀ crispants for shank3aΔN and bΔN were grown to adults, outcrossed and alleles were characterized in this F₁ generation. Once alleles were selected, the genotyping strategy (Supplementary Fig. 1) was developed. Adults with selected alleles for shan3aΔN and bΔN were then crossed to each other to generate generations doubly mutant for *shan3aΔN* and *bΔN* for experiments. We are in the process of submitting these lines to the Zebrafish International Resource Center and in the meantime can provide the lines to interested labs upon request.

**Visual motor behavior assay (VMR)**. VMR experiments were performed using a DanioVision system™ (Noldus, Wageningen, NTD) with a DanioVision observation chamber (DVOC-0040). Videos were collected at 25 fps with 1280 × 960 resolution using a Basler acA1300-60gm camera fitted with a 12 mm Megapixel lens. All DanioVision experiments were run using an ANSI SBS compatible 96 well microtiter plate. Data was collected and analyzed using the DanioVision EthoVision XT software version 11.5 (Noldus). White light for the visual motor response assay was set at 12% intensity on the high-power setting. Fish were acclimated to the observation chamber at 28 °C in the dark for at least 1 hr. VMR experiments were run using white light cycles on-off 4 times for 5 min intervals. All behavioral experiments were recorded between 11am and 3 pm, with 2–5 independent trials

per condition. Larvae were genotyped following behavioral experiments using a restriction digest assay. Larvae were anesthetized on ice prior to lysis by exposure to 20 µL 50 mM NaOH and heated to 98 °C for 20 min to harvest their genomic DNA for use as template for PCR. Regions surrounding mutations in *shank3a* and *shank3b* were amplified using GoTaq (Promega) and primers listed in Supplementary Data 1. Products were then digested using: Taq1 (present in WT; absent in *shank3aΔN* mutant), Msp1 (present in WT; absent in *shank3bΔN* mutant), Bcc1 (absent in WT; present in *shank3aΔC* mutant), and DpnII (present in WT; absent in *shank3aΔC* mutant).

**Synaptic immunohistochemistry**. Larvae were anesthetized in 0.02% tricaine, mounted in O.C.T compound (Tissue Tek Sakura Torrance CA), and gradually frozen in liquid nitrogen. Tissue blocks were then sectioned (30 µm) on a Leica cryostat, fixed in 4% formaldehyde (diluted from 16% paraformaldehyde Pierce Scientific) for 10 min and stained with synaptic antibodies as in ref. [38]. Anti-PSD-95 (1:500; Abcam; Cambridge, UK; ab-18258) and anti-Shank3ab (1:200; sc-30193, Santa Cruz Biotechnology, CA; this polyclonal has since been discontinued) were used as primary antibodies, with secondary antibodies conjugated to Alexa Fluor 568 (Abcam, ab175472) and Alexa Fluor 633 (Thermo Fisher Scientific, R21070), respectively. Images were collected using a Leica Sp6 confocal microscope, with 40x and 63X oil immersion lenses.

**Brain-wide phospho-ERK MAP-mapping**. Zebrafish were processed using a previously published protocol for comparing ERK phosphorylation or mitogen-activated protein kinase (MAP)-mapping[12]. Chemical fixation was performed using 6 well-plate baskets to quickly transfer larvae into 4% formaldehyde in 1% PBSTx (0.25%). Fixed larvae were digested with room temperature 0.25% Trypsin, incubated on ice for 45 min. Primary total ERK (tERK; 1:500) and phosphorylated-ERK (pERK; 1:500) antibodies were incubated for 3 days at 4 °C. Z-stacks of zebrafish brains were collected on a SP5 confocal microscope (Leica; Wetzlar, DE) using a dry 20x objective, with a voxel resolution of 1.5 × 1.5 × 1.98 mm. Anterior and posterior brain stacks were stitched using the Fiji pairwise stitching application[39]. Zebrafish stacks were then registered to a reference brain using CMTK[40] on the University of Miami's super computer, Pegasus. Prior to MAP-mapping, zebrafish stacks were inspected for artifacts associated with brain registration and distorted or poorly warped stacks were discarded.

To generate "lights-on" and "lights-off" maps, the Noldus DanioVision observation chamber was used with the same light settings as those used to capture VMR behaviors. For the "lights-on" condition, larvae were dark adapted for 30 min

followed by a 5-min exposure to the lights-on stimulus. For the "lights-off" condition, larvae were dark adapted for 30 min followed by lights-on for 15 min, and then lights-off for 5 min. Six larvae/basket were placed in basket inserts of six-well plates (Netwell Insert 74 μm mesh, Corning Inc., Corning, NY) so they could be rapidly transferred to fixative after the delivery of sensory stimuli.

**Cell transplantation**. All cell transplantations were completed using a CellTram Oil transplantation rig (#920002030, Eppendorf, Hamburg, DEU) and SMZ-1B dissecting scope (Nikon, Shinagawa, Tokyo, JPN). 1.0 mm borosilicate glass capillaries were pulled using a Sutter Instruments P-97 micropipette puller (Sutter, Novato, CA, USA) and broken at a 45° angle, producing a 35–45 μm inner diameter. To prevent jagged edges, the microinjection needles were then heat polished using a custom microforge.

Donor embryos were injected with a stable dextran based fluorescent dye (#D22910, Invitrogen, Waltham, MA, USA) at the one-cell stage, and both donor and recipient embryos were grown to the end of the germ ring stage (approximately 6 hpf) before dechorionation. Embryos were dechorionated using a pronase solution (1 mg/ml in system water) for approximately 6 min or until visible dechorionation was achieved. Embryos were then immediately transferred to system water, and washed 3x for 5 min. Dechorionated embryos were transferred to a penstrep solution (100 I.U./mL penicillin and 100 μg/mL streptomycin) and moved to an agarose transplantation mold in an alternating pattern of donor and recipient embryos.

Using a blunt glass pipette or the transplantation needle, donor embryos were oriented to pull cells from the top of the embryo. Approximately 40–50 cells were slowly removed from a donor embryo to prevent shearing and placed in presumptive hindbrain region of recipient embryos, based on neural fate map[16]. Recipient embryos with donor cells were then moved to a glass petri dish with system water with penstrep and allowed to grow for 24 h at 28 °C before screening. Successful transplantations were screened for brain regions of interest. Following behavioral screening, transplanted larvae were imaged to visualize donor-derived hindbrain neurons.

**Whole-brain morphometric analysis using CobraZ**. The program Advanced Normalization Tools (ANTs) was used to register zebrafish mutant and wild-type tERK image stacks collected for our previous MAP-map analysis. Using the same command script from previously published methods[19,36], mutant and wild-type tERK image stacks were registered to the Zebrafish Brain Browser (ZBB) tERK reference brain and the ZBB atlas was then back registered to all mutant and wild-type larvae, providing automated brain segmentation for all larvae. Mutant and wild-type brains were then volumetrically compared using the whole-brain morphometric analysis program Comparative Brain Analysis for Zebrafish (CobraZ)[36]. CobraZ measures and compares the volume of 180 unique neuroanatomical regions in each larvae. Additionally, we ran our updated atlas that include 13 regions with broad homology to humans, along with a 26 brain region atlas, that included molecularly and functionally unique regions (e.g. locus coeruleus). These segment volumes were then statistically compared across all genotypes in PRISM (GraphPad Software, Inc.), using a Kruskal-Wallis non-parametric ANOVA with Tukey-corrected multiple comparisons.

**Statistics and reproducibility**. With the exception of MAPmapping, data were analyzed using PRISM Version 8.1.2 (GraphPad Software Inc., San Diego, CA).

Behavioral data collected with Noldus Daniovision were analyzed using t-tests (Dunn and Bonferroni alpha corrected) and non-parametric Wilcoxon rank score test (Mann–Whitney rank scores). When there were more than two groups, a Kruskal–Wallis ANOVA test was first conducted and when $p < 0.05$, followed by a Dunn's multiple comparisons test. For CobraZ, brain segment sizes were analyzed using a non-parametric Kruskal-Wallis one-way ANOVA and followed by a Dunn's corrected multiple values comparison.

MAP-mapping was analyzed using Fiji and MATLAB scripts from the Engert lab (https://github.com/owenrandlett/Z-Brain). These scripts normalize each Z-stack by dividing pERK by total ERK (tERK), then combine groups of ERK stacks, to produce a median value for each voxel across the brain and rostral spinal cord. Median stack pERK intensity is then compared, between the first and second group, to provide a statistical difference for each voxel. Voxel z-scores that reach the cut-off of $p < 10^{-5}$ between lights-on and lights-off conditions are displayed as z and zy stack projections. Voxels $p < 10^{-5}$ between groups were then color-coded, green for group one/lights-on, and magenta for group two/lights-off. Transverse images and regional delineation of neuronal populations were created using Z-Brain reference libraries and the Z-brain viewer MATLAB application.

To achieve experimental robustness that avoids type 1 error, we generated two independent mutant models and conducted experiments on multiple batches of larvae from independent crosses for each model. Each larva sampled for a given experiment is considered a biological replicate. No statistical power calculation was conducted prior to the study and sample sizes were based on the available data. Zebrafish clutches are large and sample sizes reflect this. No data were excluded. We randomly selected individuals s from clutches of larvae for each experiment. Larvae were genotyped after experiments to ensure that links between genotype and behavior were accurate.

**Reporting summary**. Further information on research design is available in the Nature Research Reporting Summary linked to this article.

## Data availability
Brain-wide pERK/tERK immunohistochemistry stacks have been deposited at BioStudies under accession number S-BSST741.

## Code availability
MATLAB and Fiji scripts for MAP-mapping are available through the Engert lab (https://github.com/owenrandlett/Z-Brain).

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

## Acknowledgements

The authors would like to thank Drs. Sheyum Syed, Kevin Collins, James Baker, and Athula Wikramanayake for providing valuable feedback during the writing and editing process. Also, thank you to Dr. Sheyum Syed for writing custom MATLAB scripts to analyze raw behavioral data (available on request), and Drs. Owen Randlett and Harry Burgess for their help with physiological MAP-map and anatomical CobraZ analyses, respectively. To our fish facility manager, Ricardo Cepeda, thank you for keeping the fish happy and healthy. This work was supported by Bridge Funds from the College of Arts and Sciences at the University of Miami and NIH grants NIMH R03MH103857, NICHD R21HD093021, and SFARI Pilot grant 719401 to JED and an IMSD graduate fellowship from NIH parent grant R25GM076419, HHMI teaching fellowship, and McKnight Dissertation Fellowship to DMJ.

## Author contributions

R.A.K., D.M.J., and J.E.D conceived the project. R.A.K., D.M.J, S.Z., and J.E.D. wrote the manuscript. R.A.K., D.M.J., I.V., and S. H. S. conducted experiments and analyzed data. All authors contributed to the editing of the final intellectual product.

## Competing interests

The authors declare no competing interests.
