## [Transparent Peer Review File · Communications Biology]

Reviewers' comments:

Reviewer #1 (Remarks to the Author):

The authors developed two zebrafish Shank3 models to characterize the basis of sensorimotor deficits, which are a consistent feature in SHANK3 deficient human patients, manifesting ASD. Each mutant model targeted a different functional motif in Shank3, and impressively consisted of two independent mutants in the "a" and "b" gene isoforms. To assay sensorimotor function, the authors used the VMR assay, a staple in zebrafish, combined with whole-brain ERK/pERK activity mapping to identify regions of impaired function. Activity mapping demonstrated severe loss of hindbrain activity. To verify that the loss of hindbrain sensorimotor function is the basis of impaired VMR, wildtype transplants were performed targeting the hindbrain and largely restoring function. Overall, this is a well planned and concise characterization of visuomotor function in two mutant models in which the underlying genes are of importance to the broader community. Below are several comments that I believe will help to clarify the conclusions and discussion.

1. The authors show attenuated activity following the loss of illumination and use this to conclude a sensorimotor defect. However, inspecting the VMR analysis potentially suggests a reduction in baseline locomotor activity, implicating a global motor impairment versus a specific sensorimotor defect. It would be beneficial for the authors to compare light ON activity between genotypes.
2. In its current format, the Shank3 immunolabeling is challenging to resolve.
3. The authors discuss developmental defects as a basis of the observed phenotypes. To this end, it would be valuable if the authors could include representative images of wt and dual Shank3 mutants, permitting some assessment of overall morphology. Furthermore, can the authors leverage the whole brain total ERK labeling to comment on general brain morphology?
4. I feel the authors undersell their work with a limited discussion despite many exciting avenues. For example, the strength of using dual mutations to yield a robust model, potential variation in the N and C models, power of whole brain mapping to resolve phenotypes, and how hindbrain rescue is sufficient despite the potential for remaining developmental defects.

Minor

1. Figure S1a seems to be a duplicate of Figure 1A and adds no new information
2. Additional ERK mapping detail would be helpful in the methods – specifically larvae handling and sensory exposure before fixation and labeling
3. Line 55 "most common types of SHANK3 mutations..." needs a reference
4. Figure 1e should probably be referenced with other panels on Line 70
5. Sentence starting on Line 72 "...VMR deficits shank3ab/-..." needs "in"
6. Line 81 and 256. $P < 10^{-5}$ may need to be -5
7. Cannot find a reference in the text to Figure S4.
8. Line 82 reference to Figure S6 seems premature
9. Line 100 – Figure 3 reference may need to include panels e and f.
10. Line 107 – GCaMP
11. Zebrafish atlas references such as neuropil areas and similar names mean little on their own. Any reference or discussion of these regions to provide context would be immensely helpful to a general readership.
12. Line 275 – "ample sizes"
13. Line 429 neuropil misspelled

Reviewer #2 (Remarks to the Author):

In this manuscript, Kozol and colleagues use activity mapping and chimeric rescue experiments to investigate which parts of the brain require Shank3 function for normal responses to visual stimulation. Human mutations in Shank3 cause PMS, a syndromic form of ASD in which patients show hyporeactivity to light stimuli. Zebrafish Shank3 mutants also exhibit this visual hyporeactivity, making them an excellent model to learn which aspects of brain function, from light reception through motor response, require Shank 3 activity. The authors quantified behavioral changes to a light stimulation using a previously established behavioral assay and found that Shank3 homozygous mutants responded less well than heterozygotes which responded less well than wildtypes. They then mapped brain activity during the behavioral task using pERK and found that regions involved in light reception showed normal activity, whereas hindbrain regions involved in behavioral responses showed muted activity. To learn which of these areas required normal Shank3 function, the authors generated chimeric animals by transplanting wild type cells into regions destined to develop as hindbrain. They identified two regions – vGluTcluster 2 and Neuropil region 4 – as brain areas that supported normal behavior when these cells were derived from wild-type donors, even though the rest of the animal was mutant, showing that these brain regions specifically require Shank3 function for normal responses to light. These results provide evidence that activity dysfunction in these hindbrain cells plays an important role in the behavioral deficits of Shank3 mutants in response to light stimulation.

This is a very nice piece of work that provides insight into where Shank3 activity is required for normal behavioral responses to light stimulation. The behavior, activity mapping and transplantation experiments are beautifully done and very well documented, and the results are robust. Knowing which brain regions are affected by loss of Shank3 function is a significant step forward in understanding PMS. However, this work does not entirely support the authors' conclusion that they have identified the neurobiological basis of sensory hyporeactivity in Shank3 loss-of-function zebrafish models of PMS. Such an understanding requires establishing more of a link between the underlying processes within the affected brain regions and sensory hyporesponsiveness. For example, the authors suggest that synaptic defects and/or developmental delays could be the cause of the brainstem deficits they observe. Testing these possibilities would provide important insights into the processes that fail in the hindbrain nuclei of Shank3 mutants, pinpointing the cellular mechanisms that require Shank3 activity for normal sensorimotor integration, thus increasing the impact of this work. For example, the authors could examine morphology of the nuclei they identified to determine whether the neurons, or adjacent non-neuronal cells appeared underdeveloped in Shank3 mutants. They could use immunohistochemistry, in situ hybridization, or other techniques to examine synapses directly to see whether they differ in number, morphology, pre- or post-synaptic proteins, or other parameters from those of wildtypes.

Author response to reviewers→

original reviewer comments in italics; author responses in green regular text.

Please note that all changed text in the PDF document is in red.

Referee #1: zebrafish models of neurodevelopmental disease, neuroimaging, behavior

Referee #2: zebrafish neuroscience and neurological disease

Reviewers' comments:

Reviewer #1 (Remarks to the Author):

The authors developed two zebrafish Shank3 models to characterize the basis of sensorimotor deficits, which are a consistent feature in SHANK3 deficient human patients, manifesting ASD. Each mutant model targeted a different functional motif in Shank3, and impressively consisted of two independent mutants in the “a” and “b” gene isoforms. To assay sensorimotor function, the authors used the VMR assay, a staple in zebrafish, combined with whole-brain ERK/pERK activity mapping to identify regions of impaired function. Activity mapping demonstrated severe loss of hindbrain activity. To verify that the loss of hindbrain sensorimotor function is the basis of impaired VMR, wildtype transplants were performed targeting the hindbrain and largely restoring function. Overall, this is a well-planned and concise characterization of visuomotor function in two mutant models in which the underlying genes are of importance to the broader community. Below are several comments that I believe will help to clarify the conclusions and discussion.

1. The authors show attenuated activity following the loss of illumination and use this to conclude a sensorimotor defect. However, inspecting the VMR analysis potentially suggests a reduction in baseline locomotor activity, implicating a global motor impairment versus a specific sensorimotor defect. It would be beneficial for the authors to compare light ON activity between genotypes.

Thank you for your observation. We have added this analysis to Supplementary Fig. S3, Tables. S9-14 (Text Lines 72-74; Fig. S3 Lines 858-870; Tables S9-14 Lines 1109-1177). Swimming in the lights-on phase is more variable in our hands between individuals and batches than that in the lights-off phase, for example see variation between wild type in different experiments, but we agree that this quantification is important to show.

2. In its current format, the Shank3 immunolabeling is challenging to resolve.

We agree that that the size of the puncta makes them challenging to see in most panels. To address this we have changed the color, increased the magnification in the supplementary figures, and added arrowheads.

Wildtype Shank3 expression in Cerebellum (Ce) and Medulla Oblongata (MO)

3. The authors discuss developmental defects as a basis of the observed phenotypes. To this end, it would be valuable if the authors could include representative images of wt and dual Shank3 mutants, permitting some assessment of overall morphology. Furthermore, can the authors leverage the whole brain total ERK labeling to comment on general brain morphology?

This is a great suggestion. We have leveraged tERK staining to compare the morphology of *shank3ab* mutant and WT brains. See Text (Lines 204-214, 242-243, & 1369-1384), Figures 4 (Lines 484-492), Supplementary Figure 9 (Lines 992-1102), and Supplementary Tables 33-41 (Lines 1235-1331).

4. I feel the authors undersell their work with a limited discussion despite many exciting avenues. For example, the strength of using dual mutations to yield a robust model, potential variation in the N and C models, power of whole brain mapping to resolve phenotypes, and how hindbrain rescue is sufficient despite the potential for remaining developmental defects.

Thank you for this suggestion. We have added a paragraph before the conclusions section that points to the strength of our experimental approach for yielding new insights (Lines 218-230).

“In summary, we used whole-brain activity mapping and chimeric rescue experiments to identify the rostral brainstem as a region that requires Shank3 function to generate behavioral responses to visual stimuli. To do this we generated two independent zebrafish *shank3abΔN* and *shank3abΔC* mutant models of Phelan McDermid Syndrome, both of which, like humans with Phelan McDermid Syndrome, showed dampened responses to visual stimuli. Unbiased, whole-brain activity mapping in both *shank3ab* models were consistent, demonstrating that regions of the brain that detect light were activated normally while brain regions that integrate sensory information to produce specific motor responses showed reduced activation. Restoring Shank3 in brainstem nuclei by transplanting WT cells into *shank3ab* mutant embryos at gastrula stages was sufficient to rescue dampened larval sensory responses, demonstrating a critical role of Shank3 for integrating sensory information in the rostral brainstem. An analysis of brain regions that could rescue sensorimotor behaviors in *shank3ab* models supports an essential role for *shank3* in these cerebellar-like brainstem circuits.”

Minor

1. Figure S1a seems to be a duplicate of Figure 1A and adds no new information

Yes, we have removed panel 1A.

2. Additional ERK mapping detail would be helpful in the methods – specifically larvae handling and sensory exposure before fixation and labeling

Thank you for pointing out that this description was lacking. Lines 1277-1284 We have added the following description of larval handling pre-fix.

“To generate ‘lights-on’ and ‘lights-off’ maps, the Noldus DanioVision observation chamber was used with the same light settings as those used to capture VMR behaviors. For the ‘lights-on’ condition, larvae were dark-adapted for 30 minutes followed by a five minute exposure to lights-on stimuli. For the ‘lights-off’ condition, larvae were dark adapted for 30 minutes followed by lights-on for fifteen minutes, and then lights-off for five minutes. Six larvae/basket were placed in basket inserts of six-well plates (Costar Netwell Insert 74μm mesh) so they could be rapidly transferred to fixative after the delivery of sensory stimuli.”

3. Line 55 “most common types of SHANK3 mutations...” needs a reference

Reference has been added (Line 55).

4. Figure 1e should probably be referenced with other panels on Line 70

Yes, good point → we added 1e here (Line 70).

5. Sentence starting on Line 72 “...VMR deficits *shank3ab*-/-...” needs “in”

“in” has been added (Line 74).

6. Line 81 and 256. $P < 10^{-5}$ may need to be -5

Thank you for catching that → now fixed (Line 83).

7. Cannot find a reference in the text to Figure S4.

Thank you for catching that → a reference to S4, now S6, has been added to the reference on line 106 (Line 104).

8. Line 82 reference to Figure S6 seems premature

Agreed, that reference has been removed.

9. Line 100 – Figure 3 reference may need to include panels e and f.

Line 103 Changed from ‘3b-d’ to ‘3b-f’.

10. Line 107 – GCaMP

Line 124 This has been fixed.

11. Zebrafish atlas references such as neuropil areas and similar names mean little on their own. Any reference or discussion of these regions to provide context would be immensely helpful to a general readership.

Thank you for this recommendation. We have added two paragraphs dedicated to this issue that integrate functional and structural studies in both zebrafish and mammals (Lines 118-141).

12. Line 275 – “ample sizes”

Line 320 ☺ “VMR line graphs with ample sizes are indicated...” has been corrected to read “VMR line graphs with sample sizes indicated...”

13. Line 429 neuropil misspelled

Line 611 This has been fixed.

Reviewer #2 (Remarks to the Author):

In this manuscript, Kozol and colleagues use activity mapping and chimeric rescue experiments to investigate which parts of the brain require Shank3 function for normal responses to visual stimulation. Human mutations in Shank3 cause PMS, a syndromic form of ASD in which patients show hyporeactivity to light stimuli. Zebrafish Shank3 mutants also exhibit this visual hyporeactivity, making them an excellent model to learn which aspects of brain function, from light reception through motor response, require Shank 3 activity. The authors quantified behavioral changes to a light stimulation using a previously established behavioral assay and found that Shank3 homozygous mutants responded less well than heterozygotes which responded less well than wildtypes. They then mapped brain activity during the behavioral task using pERK and found that regions involved in light reception showed normal activity,

whereas hindbrain regions involved in behavioral responses showed muted activity. To learn which of these areas required normal Shank3 function, the authors generated chimeric animals by transplanting wild type cells into regions destined to develop as hindbrain. They identified two regions – vGluT cluster 2 and Neuropil region 4 – as brain areas that supported normal behavior when these cells were derived from wild-type donors, even though the rest of the animal was mutant, showing that these brain regions specifically require Shank3 function for normal responses to light. These results provide evidence that activity dysfunction in these hindbrain cells plays an important role in the behavioral deficits of Shank3 mutants in response to light stimulation.

Thank you for this excellent summary.

This is a very nice piece of work that provides insight into where Shank3 activity is required for normal behavioral responses to light stimulation. The behavior, activity mapping and transplantation experiments are beautifully done and very well documented, and the results are robust. Knowing which brain regions are affected by loss of Shank3 function is a significant step forward in understanding PMS. However, this work does not entirely support the authors' conclusion that they have identified the neurobiological basis of sensory hyporeactivity in Shank3 loss-of-function zebrafish models of PMS. Such an understanding requires establishing more of a link between the underlying processes within the affected brain regions and sensory hyporesponsiveness. For example, the authors suggest that synaptic defects and/or developmental delays could be the cause of the brainstem deficits they observe. Testing these possibilities would provide important insights into the processes that fail in the hindbrain nuclei of Shank3 mutants, pinpointing the cellular mechanisms that require Shank3 activity for normal sensorimotor integration, thus increasing the impact of this work. For example, the authors could examine morphology of the nuclei they identified to determine whether the neurons, or adjacent non-neuronal cells appeared underdeveloped in Shank3 mutants. They could use immunohistochemistry, in situ hybridization, or other techniques to examine synapses directly to see whether they differ in number, morphology, pre- or post-synaptic proteins, or other parameters from those of wildtypes.

Underline 1: Thank you for your very accurate assessment. We have reworded the last sentence of the introduction to read “Here we identify rostral brainstem as a region that requires Shank3 function for normal behavioral responses to light stimulation in zebrafish models of PMS.”

Underline 2: This is a great suggestion. We have leveraged tERK staining to compare the morphology of *shank3ab* mutant and WT brains. See Text (Lines 204-214, 242-243, & 1369-1384), Figures 4 (Lines 484-492), Supplementary Figure 9 (Lines 992-1102), and Supplementary Tables 33-41 (Lines 1235-1331).

See figures embedded above in response to a similar point from reviewer 1.

REVIEWERS' COMMENTS:

Reviewer #1 (Remarks to the Author):

The authors added a significant amount of new analysis and clarification. Leveraging available brain atlases was especially a powerful addition to resolve neural substrates for sensorimotor control; focusing their analysis and discussion. All of my concerns have been addressed. This is a nice study highlighting the utility of zebrafish as a disease model and should be of interest to a broad readership.

Reviewer #3 (Remarks to the Author):

The authors have demonstrated that the shank3 transplant rescue experiments alter the size defects of two brain areas that are altered in shank3 mutants, particularly the hindbrain, which they have shown is the source of VMR defects. This goes a long way to resolving the concerns of Reviewer 2 in favor of a developmental hypothesis, which is also well contextualized in the manuscript. This is not 100% conclusive-- it could be there are defects in *both* development/growth and synapse number/maturity in the hindbrain. But I believe this is sufficient evidence in support of a developmental effect, esp. given that resolving synaptic size/number/density in the hindbrain would be experimentally non-trivial and difficult to distinguish from the size/development phenotype.

Overall, I am impressed with this really cool paper, which makes great use of mutants, behavior, whole-brain activity mapping, and a really satisfying transplant hindbrain rescue experiment.